# Urban Regeneration Involving Communication between University Students and Residents: A Case Study on the Student Village Design Project

**DOI:** 10.3390/ijerph192315834

**Published:** 2022-11-28

**Authors:** Joo Young Kim, Jung Hoon Kim

**Affiliations:** 1Department of Architecture, Sejong University, Seoul 05006, Republic of Korea; 2Smart City Research Center, Korea Institute of Civil Engineering and Building Technology, Goyang-si 10223, Republic of Korea

**Keywords:** urban regeneration, community planning, campus town, resident participation

## Abstract

This study analyzed the effect of the application method of community planning in the case of the Student Village Design Project. Urban regeneration is a method that develops a city with the direct involvement of residents. However, in Korea, urban renewal projects are focused on external expansion, thereby generating conflicts among residents and prolonging these projects. In particular, university towns are experiencing various types of conflicts compared with other urban regeneration projects because the lives of residents and university students are overlapped. Therefore, the research method is conducted as follows to analyze the communication effect in accordance with the purpose of the study. First, we reviewed community planning methods as led by university students. Second, we applied community planning to the project. Third, the results and effects of resident communication were analyzed after applying community planning. Fourth, in the student-led urban regeneration, a community planning method that has results and effects in resident communication was derived. We found that community planning is a significant means of communication between university students and residents. The concrete finding was derived from necessary and optional methods that have high communication effects with residents among the community planning methods.

## 1. Introduction

### 1.1. Research Background

The Korean economy is transitioning from high to low growth. City management projects in this low-growth period are primarily focused on improving the quality of citizen life, rather than on external expansion and growth [1]. Among them, the urban regeneration method, which utilizes local community space to induce social exchanges among residents, is receiving increasing attention as a city management strategy [2]. The Seoul Metropolitan Government began various urban regeneration projects, to improve the local residential environment in underdeveloped areas around its university towns, as of 2011. However, these urban regeneration projects involving universities tended to focus on physical facility extension (e.g., land purchase, expansion of university boundaries), which then increased land prices in adjacent sites, and generated conflicts with local communities through problems such as landscape and natural environment destruction, in addition to being inadequate as a city management technique in the low-growth period [3]. Thus, the Seoul Metropolitan Government is carrying forward the “Campus Town Project” as a city management technique in this low-growth period, in order to help universities to coexist with their wider communities [4,5]. Prior research related to housing regeneration in university towns has found that constant discussion between residents and universities leads to higher satisfaction, following such development [6,7,8]. However, most housing regeneration projects in university towns are carried out directly by the universities [4,9,10,11,12,13], with relatively few cases in which university students communicate with residents; there are also few studies on community planning techniques in this context.

Thus, the current study derived effective community planning methods for urban regeneration by organizing the process in which university students communicate with residents, and analyzed their performance and outcomes. Moreover, as public design (including the street environment, community, and infrastructure) appears to be an important factor in urban regeneration projects [14,15], this study focused on the public design of urban regeneration. To this end, it analyzed a specific case (the Student Village Design Project) in which University A conducted a public design project with university students aimed at urban regeneration. The spatial scope was the residential area around University A, and the temporal scope was from June 2020 to January 2021, during which the Student Village Design Project was implemented.

The main objectives and methods of this research are summarized as follows. First, community planning methods through which students and residents can communicate are reviewed, and previous studies on the importance and methods of urban regeneration that involves resident participation. Second, this study provides information on how to apply community planning methods to the Student Village Design Project. Third, we analyze the effects of community planning methods in each phase of the Student Village Design Project. Finally, we derive and discuss community planning methods for urban regeneration, in which students and residents can communicate better.

### 1.2. Importance of Resident Participation in Urban Regeneration

Urban regeneration is not planned by either administrators or policymakers, but is rather conducted democratically through involving citizens in the planning process [16]. The role of resident participation in urban regeneration is to exert influence over the process of shaping, selecting, and executing public policies [17,18,19]. In this process, public concerns, needs, and values are important factors within public decision making [20]. In other words, a two-way communication and interaction process between residents and the public sector involves establishing policies that can be both supported and accepted by residents [21].

Some notable finding on the effects of resident participation on urban regeneration are as follows. First, resident participation in urban regeneration alleviates social disadvantages and alienation [22,23,24,25,26]. Moreover, the rationality and democracy of a given plan can be secured by involving residents in the processes of planning and implementation [27]. Hong [12] argues that residents with interests in urban regeneration can find ways to resolve or minimize conflicts, by directly participating in these projects. This is because these residents are most acutely aware of the local regional characteristics. Further, Elster [28] and Jin et al. [29] revealed that resident participation has a positive effect on neighborhood satisfaction. This result is related to the fact that residents’ community spirit is subsequently strengthened, and ultimately contributes to regional revitalization. However, resident participation also has some negative effects. For example, Ferilli et al. [30] explain that resident participation is critical in social storytelling, community informatics, relational public art, and culture projects, but that it may also instigate inequality by investing power with only a few people. Furthermore, Hong [12] argues that resident participation unnecessarily extends the discussion period, due to the resulting complexity with stakeholders. Thus, it is important to have appropriate regulations on the extent and methods of stakeholder involvement.

## 2. Material and Methods

### 2.1. Research Data

#### 2.1.1. Data of the Student Village Design Project

University A carried out the Student Village Design Project as part of Seoul Metropolitan Government’s Campus Town Project, using the results of this study. The data used included the planning process, execution process, results, and student satisfaction survey after the project. The Student Village Design Project was carried out across two phases, with the first phase taking place from June to December 2020, while the second was from January to June 2021, thereby lasting six months each.

#### 2.1.2. Study Area

The study site is the area represented by the red dotted line in Figure 1. This area is approximately 106,450 m^2^, and consists of 65% detached houses, 20% neighborhood living facilities, 11% apartment houses, and 4% other buildings [31]. The section where the red dotted line and the blue solid line in Figure 1 overlap is the entrance to the university, which is surrounded by retaining walls. Neighborhood commercial facilities (highlighted in yellow in Figure 1) are located around the main road, on the west and south sides. This site is used as a passage for students to access neighborhood commercial facilities. This site has a system in which a resident council is formed to discuss the neighborhood’s issues, led by a resident representative. There are certain problems, such as a lack of parking space, narrow roads, noise problems, insufficient infrastructure, and security issues. Moreover, the residential space of citizens and the activity space of university students overlap, meaning that there are trash disposal and noise issues caused by students [32].

#### 2.1.3. Participant Characteristics

In total, 60 university students participated in the Student Village Design Project: 36 in the first phase and 24 in the second. There were 46 students from the department of architecture, four from the department of industrial design, and 10 from various other departments, such as design innovation, information protection, tourism and hospitality, urban sociology, and painting. These university students were leaders of the Student Village Design Project, with this study referring to them as “Student Village Designers”. There are approximately 3000 residents living in the target area, and the average age of residents is 41 years [33]. The Student Village Design Project was carried out based on individual communication between students and residents, with some key matters being directly discussed with the resident representative.

### 2.2. Research Methods

#### 2.2.1. Research Process

First, this study explained the Student Village Design Project in each stage of its process, and analyzed the activity phase and application methods of the relevant community-based planning. The process was conducted in the order of a basic survey and setting, selecting student village designers, workshops, preparing and supplementing the action plan, and project implementation. The basic survey involved conducting a situation survey, deriving key issues, and planning publicity ideas. The process of selecting Student Village Designers included putting up a recruitment notice, assessment, and a launching ceremony following selection. Steps taken in the workshop included deriving regional agendas (topics), finding problems and their causes, deriving implementation methods, and design conception, with it being carried out four times. Preparing the action plan included basing it on the workshop’s results, a discussion with related agencies, and revising it through expert mentoring. Project implementation involved construction and holding an awards ceremony, and a performance-sharing event. Second, this study examined the effects of each community planning method, with a focus on resident communication and outcomes. Third, this research investigated and discussed effective urban regeneration methods, in which students and residents communicated.

#### 2.2.2. Process and Methods of Urban Regeneration That Involve Resident Participation 

Wates [34] outlines various plans involving resident participation across 53 methods. Among them, 37 can be applied to the process in which university students and residents communicate to create public design. This is summarized in Table 1. The methods in Table 1 can be classified into planning, discussion, survey, interactive display, publicity, assessment, creating and using space, creating and using organizations, and budget support. 

Each method can be described as follows. Planning involves participants discussing the content, period, and stages of planning, including action planning events and briefing workshops. Discussion involves finding agendas and deriving ideas from various perspectives, and includes activity weeks, art workshops, design games, design workshops, gaming, planning days, practical planning, planning weekends, prioritizing, process planning sessions, reconnaissance trips, photo surveys, risk assessments, and simulations. Interactive display is a stage that involves communicating and discussing opinions with residents, while also involving wider publicity. Surveying includes choice catalogues, community profiling, and table scheme displays. Additionally, the visual expression of a situation analysis, and the relevant discussion results, can be presented with diagrams, electronic maps, elevation montage, mapping, and models. The assessment’s purpose is to develop creative ideas and increase initiative, involving award schemes, idea competitions, and review sessions. Publicity involves publicizing the project content, ideas, and outcomes, which includes design fests, local design statements, newspaper supplements, open house events, street stalls, and video soapboxes. Creating and using space involves the development of a physical environment for urban regeneration participants to gather and share information, including architecture and community design centers. Finally, organizations include design assistance teams, planning aid schemes, and user groups.

## 3. Analyzing the Effects of Student Village Design According to the Community Planning Method

### 3.1. Analyzing Each Stage of the Community Planning Method

#### 3.1.1. Basic Survey

A basic survey is the process of discovering key regional issues through a situation analysis of the study site, as well as making plans for publicity. A situation survey was conducted in the form of a resident survey, to investigate the sites that required improvement in the local residential environment. The respondents were to mark these areas on the map, and explain their reasoning for the same. The survey was conducted face to face, with 74 copies of the questionnaire being used. As a result, the respondents marked, on the map, areas that require sanitary management, areas with frequent noise issues, and areas that require environmental improvement. According to Park and Yang [35], it was revealed that university towns have noise and poor sanitation, which provides a reliable basis for the survey results. This kind of resident survey is used to conduct investigations for planning, as shown in Table 2, and applied “community profiling”.

The survey results can be mapped as follows. There were 26 spots and five streets that required improvement, in the residential environment in question. Sites with illegal trash disposals are marked using blue dots, those with safety issues are marked using gray dots, streets with frequent motorcycle travel are marked with orange lines, sites with numerous illegally parked cars are marked with purple lines, and those with noise issues are marked using green dots (Figure 2).

Publicity planning was conducted to recruit Student Village Designers, and communicate with residents. Publicity applies the “community design center” method, using the Campus Town support center as its key base. A banner showing a QR code is attached in front of the support center, displaying the project’s details and information (Figure 3a), with a resident opinion box installed in the lobby of the support center, to gather resident opinions (Figure 3b). Moreover, a YouTube channel was set up for publicity, and the collection of resident opinions (Figure 3c). This is known as “interactive display”, in the field of publicity, as shown in Table 2.

#### 3.1.2. Selecting Student Village Designers

Student Village Designers were recruited through university posters, the websites of related agencies, social media, and a YouTube channel shared among students of universities in Seoul, other than University A. The applicants were to submit their ideas through their applications for becoming Student Village Designers. Then, a launching ceremony was held to explain the meaning, roles, importance, and plans of the Student Village Design Project. This led to a “briefing workshop”, within the planning stage of community planning, as shown in Table 2.

#### 3.1.3. Workshops

Workshops were held four times, once per week, over the course of a month for Student Village Designers to investigate, discuss, and derive feasible projects (Table 3).

Regional agendas were derived during the first workshop. “Rich Picture” [36] was used in this process, to develop various agendas by drawing pictures when people with different perspectives explored complex issues (Figure 4). The visual elements of these pictures helped us to understand the connections of the parts of the project, as well as the whole [37]. This is the discussion stage in the form of an “art workshop”. The sites were selected with reference to the places where the regional problems were reported to occur in the planning stage (Figure 2).

The second workshop involved sharing the team-based survey results over the course of a week, and identifying the regional problems. For the survey, each team listed the region’s problems on a panel, and asked the residents certain questions, as they were passing by, by attaching stickers to what they thought were important topics (Figure 5a). This is the survey stage, that involves applying the “choice catalogue”. Next, a “persona-based scenario method” was used, to recreate an actual experience of the regional problems of residents. This method (developed by Alan Cooper [38]) is applied to the user-centered design. Scenarios are classified by purpose and content into context scenarios, key path scenarios, and validation scenarios. Context scenarios set the status or requirements of study sites before planning, and is suitable for application in the situation survey stage of the Student Village Design. According to the scenario, students became residents to think about the problems from three perspectives—the environmental setting, actors, and actions involved—and then gave presentations (Figure 5b,c). This process applied the “simulation” method of the discussion stage. 

The third workshop involved developing potential solutions. Solutions were derived (Figure 6a) by listing the causes of each problem, and writing how to solve them on post-its. Then, a graph was prepared, as shown in Figure 6b, to identify effective solutions. The Y-axis shows time and budget, while the X-axis shows feasibility. By rearranging the post-its from Figure 6a on this graph, effective solutions were derived. The impact–effort matrix program was used, a tool to select highly effective ideas with little effort [39]. Tavares et al. [40] argued that this tool is effective in solving urban problems. For the selected solutions, specific ideas are derived, as shown in Figure 6c. This is “planning for real” in the community planning method.

The fourth workshop involved shaping a concrete design. The project, with the ideas that were generated, was modeled according to the scale of a real residential environment (Figure 7). The ideas were then revised and concretized by imagining what it would look like after implementation. This is the “design workshop” phase in the discussion stage. Next, these results received additional critiques from outside experts. This is the “review session” in the assessment stage. 

#### 3.1.4. Preparing the Action Plan

The content of the project then needed to be concretized in order to implement it, as shown in Table 4. Accordingly, the action plan needed to include the project’s name, purpose, needs, content, required period, anticipated effect, budget plan, site photos and location map, and reference images, based on the workshop results. Next, the action plan was later supplemented through evaluation and mentoring, which was carried out by the Campus Town Project manager, a resident representative, and various designers. This can be described as creating and using organizations, using the “design assistance team”. Each team held discussions with residents (including resident briefings, opinion polls, obtaining consent) and public institutions (including related department briefing, sending official documents) about the project sites and designs. This can be described as creating and using organizations in the form of communicating with the “user group”.

#### 3.1.5. Project Implementation

The project implementation stage directly involved students in the construction process (Figure 8a). Name plates containing the list of team members who participated in the project were attached to each of the completed sites. Further, an event was held on site, in which the project was explained to residents and all other parties concerned, and then assessed. This is known as the publicity stage, in the form of “street stalls”. Later, winning works were selected based on their individual excellence, after an assessment. This applied the “award scheme” method in the assessment stage. In addition, a video outlining the goals of the Student Village Design, explanations of each project, and reviews and outcomes of the students was developed and released online (Figure 8b), which was then further publicized by the press (Figure 8c). This is, again, the publicity stage, using the “video soapbox” and “newspaper supplement” methods. The analysis results of project implementation are show in Table 4.

### 3.2. Effects of Community Planning Methods

The effect analysis of the community planning methods can be seen in terms of the effectiveness of resident communication and results-based effects; the findings are shown in Table 5. First, the effect of the community planning method differs in the result of resident communication depending on the process-based application. It is classified into direct and indirect effects. The direct effects are the results which reflect the opinions of residents directly when applying the community planning method. The indirect effects are the results decided on the matter by referring the opinions of residents. The result shows that the community planning used for direct effect is the community profiling in the basic survey stage and the interactive display in the setting stage. In the workshop, the direct effect was shown in the choice catalog method for setting problems and in the action plan, the method using the user group shows a direct effect. This was effective in checking the results that students had agonized over and worked on. The indirect effect was shown as the method of utilizing the community profiling and community design center in the basic survey stage. In the workshop, the art workshop, the simulation and the practical planning showed their effectiveness and it was also effective when local residents were included as the design assistance team in the action plan. Besides that, the indirect effect occurred in the street stall, the award scheme, and newspaper supplement of the project implementation stage. It seemed that indirect effects appeared in the process of agonizing over the local problems, deriving solutions, and expanding result. While the method of the design workshop and the review session were ineffective because of the lack of residents’ communication, it suggested that residents’ communication was important in the process and method when moving on to the next stage by experiencing a lot of mistakes. 

The effects of the research outcomes vary in terms of the required and selective process. First, an effect resulted from the process of implementing the project. Further, the steps of community profiling and mapping in the basic survey and briefing workshops, when selecting the Student Village Designers, were essential for the workshop stage. Additionally, choice catalogues, practical planning, and the design workshops that were used to identify regional problems were important in determining ideas for the project. In particular, the user groups that supplement the action plans were crucial in the development of this project. Second, a selective process that has high applicability was used. Online publicity and interactive displays are more effective than offline publicity, but are not key in determining the project. Art workshops in deriving regional agendas, regional problem simulations, review sessions in design conception, and action plan mentoring, as well as the use of the design assistance team, are all creative activities that helped to facilitate an understanding of resident life that in turn helped to improve the overall design. Street stalls, award schemes, and newspaper supplements in the project’s implementation helped to recover resident trust, derive excellent ideas, and generate effective project publicity and diffusion.

In summary, community planning requires factors such as community profiling via a situation survey, using a choice catalogue in finding regional problems, and the use of user groups, in discussion with other key parties who are concerned with preparing the action plan. Table 5 summarizes the process of resident communication, and the resulting effects, according to community planning methods, in the implementation of the Student Village Design Project. 

## 4. Discussion

This study examined effective community planning methods for urban regeneration projects by analyzing the process and performance of urban regeneration through an investigation into the communication between selected university students and residents. The study subject here was the Student Village Design Project of University A. This paper has been developed based on a literature review, three main results derived from the study, and the implications of the findings.

First, we found that the resident communication process is critical in determining the action plan as structured, in the order of publicity, opinion gathering, and the identification of regional problems. The effective communication technique of each process is community profiling for situation surveys, the interactive display of online communication, choice catalogues in identifying regional problems, and the utilization of user group discussions with all parties concerned. For resident participation, the user group method was found to be most active, followed by interactive display, community profiling, and choice catalogue. This result shows that resident participation is important for both project planning and implementation, as claimed by Elster [28] and Bae [27] et al. Moreover, this supports the findings of Hong [12], who states that adequate resident participation is only needed depending on the scope or method used, rather than participation in the entire process. In this context, active resident participation is more important in the implementation process than during planning.

Second, methods important for project implementation that do not involve communication with residents include mapping to derive key issues, briefing workshops to select project participants, practical planning to derive solutions, and design workshops and models for design conception. Mapping, briefing workshops, and practical planning were conducted with reference to the results of the resident opinion poll, which took place before these decisions. Thus, we ensured the facilitation of some degree of resident communication. Here, resident participation can cause problems such as increasing resident fatigue and delaying the project [41,42] and, as such, this study’s results are a likely solution to these problems. In addition, we found design conception to be an important method throughout this process, but that it may require resident communication further down the road. Therefore, among community planning methods, choice catalogues for design selection, table scheme displays for design exhibition, and opinion gathering must be conducted through resident communication.

Third, we found that the methods that are effective in resident communication (although they are not required) are the use of design assistance teams, street stalls, award schemes, newspaper supplements, and video soapboxes. Design assistance teams and award schemes are particularly effective in deriving designs preferred by residents. In particular, street stalls positively influenced the residents’ negative perceptions about urban regeneration and the university. This proves that the joint participation of university students and residents is likely to increase the effect of urban regeneration projects in university towns. On the other hand, we found that using a community design center had little effect in resident communication, which could be related to the restricted face-to-face activities due to the COVID-19 pandemic.

This study applies the community planning method, which several scholars recommend, and analyzes its effect. The studies of Tavares et al. [40] and Loh [43] use a similar methodology to this study. Tavares et al. [40] derived an effective community participatory method for urban design from the empirical case. Loh [43] drew a step where there may be a disconnection with various stakeholders in policy planning. Thus, this study methodology can be said to be suitable for research which analyzes the effect of applying the theoretical planning method to the empirical case.

## 5. Conclusions

This study analyzed the effect of applying the community method to the case of the Student Village Design Project to derive an effective method of urban regeneration in which students and residents communicate. In consequence, both direct and indirect methods of residents’ communication were founded, and essential and useful methods were selected in terms of results. In summary, urban regeneration projects in which university students communicate with local residents show different levels of resident communication, or outcomes/effects, depending on the community planning methods used. In other words, it was more important for residents to participate in the essential stages in an effective way than to participate in all stages. For example, the method which showed a direct effect on communication between students and residents, and was essential for effective results was shown to be community profiling in the basic survey, interactive display in online PR, choice catalog in finding regional problems, and the use of the user group in the action plan. 

This study has the following limitations. First, only 37 out of 53 community planning methods were used. Thus, the effect analysis was conducted on only these 37 methods that were applied to a real-world case, meaning that the adequacy of each method was determined in this specific context. Second, the Student Village Design Project was short-term in nature; hence, it was difficult to assess the continuity and sustainability of the observed effects. Third, the resident opinion poll was conducted on only a subset of all the local residents. This is because face-to-face contact was limited due to COVID-19, and because the project was carried out over a short period. In future urban regeneration projects, it is therefore necessary to continuously conduct research that applies various community planning methods and validates the noted effects. In addition, applying scientific research methodology, it is necessary to derive the objective and generalized research result. The results of this study can be used as basic data for establishing resident participation models and community planning methods in student-led urban regeneration projects.

## Figures and Tables

**Figure 1 ijerph-19-15834-f001:**
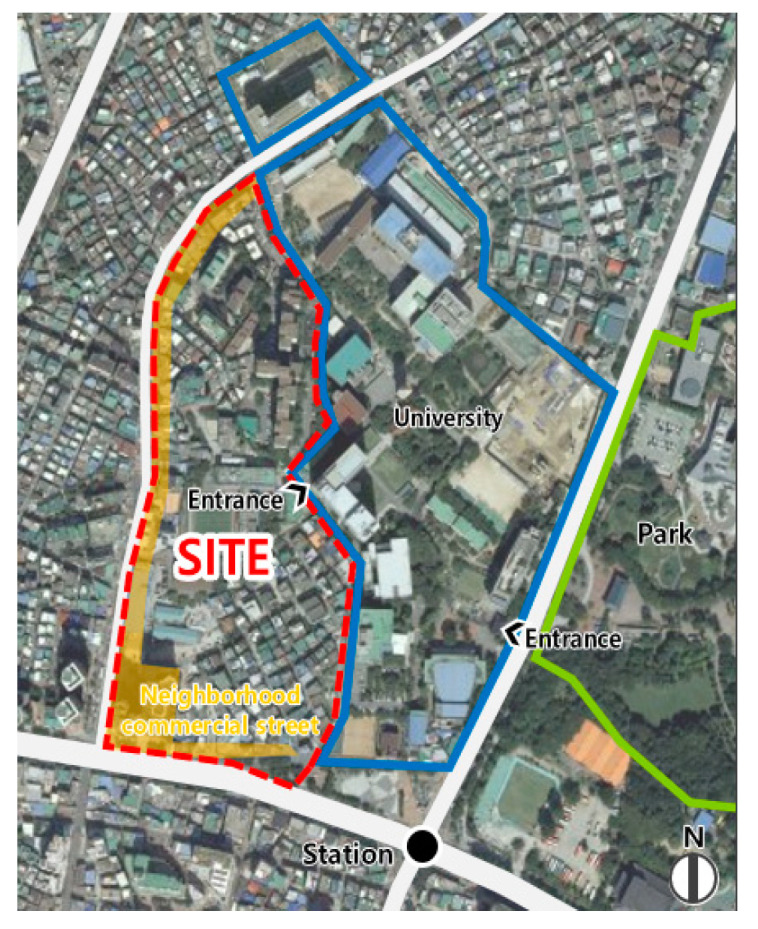
Study area.

**Figure 2 ijerph-19-15834-f002:**
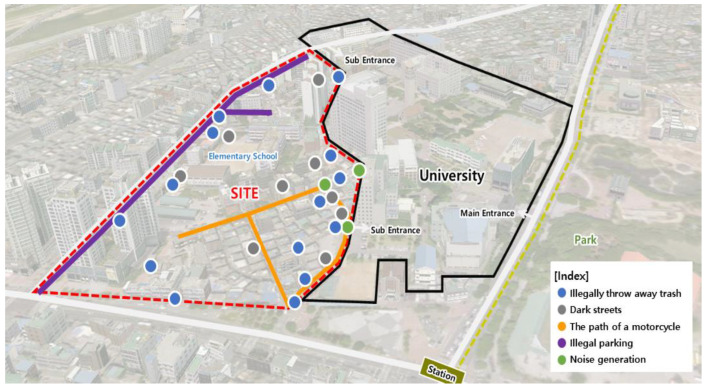
Survey results of the study sites and their regional problems.

**Figure 3 ijerph-19-15834-f003:**
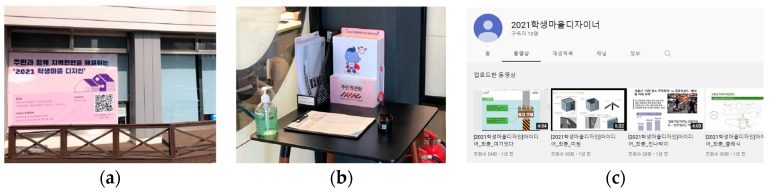
Interactive display used for publicity: (**a**) The banner in front of the main building guided the schedule and contact information under the theme of ‘2021 Student Village Design to solve regional problems with residents’; (**b**) We put a box with ‘Resident opinion box’ written on it in the lobby; (**c**) It is a YouTube channel screen, and each team’s work process was produced and uploaded as a video. An example of a video title is [2021 Student Village Design] Idea_Final_Team name.

**Figure 4 ijerph-19-15834-f004:**
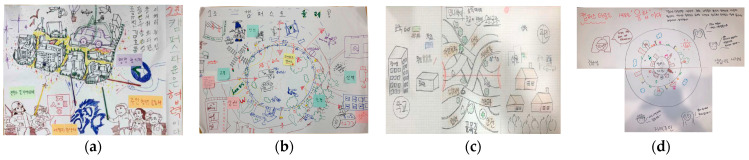
Results of detailed picture activity: (**a**) Topic is “Campus Town involves collaboration.” It means that universities, experts, administration, and local communities work together to create a village. (**b**) Topic is “Want to come to campus?” It means a town where safety, transportation, commerce, and landscape environment are created together. (**c**) Topic is “A three-legged race of university and residents in the alley.” It means universities and residents who pursue friendliness, coexistence, communication, and collaboration in alleyways. (**d**) Topic is “Campus Town as a new convergence”. It represents the students, locals, and merchants who bring vitality to a declining town.

**Figure 5 ijerph-19-15834-f005:**
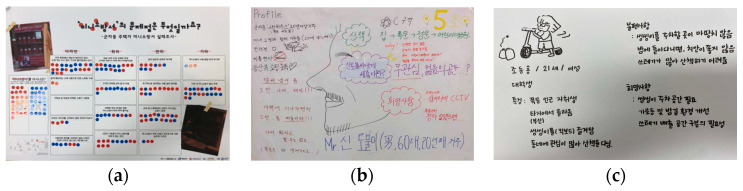
Resident opinion poll (**a**) and results of the persona method (**b**,**c**): (**a**) Stickers on problems that residents considered serious among the design, location, and management of the fire extinguisher cabinets in the village. (**b**) A man in his 60s who had lived in the village for 20 years complained about the smell of cigarettes and noises when taking walks around the university, and explained the need to set up a CCTV camera. (**c**) A female student aged 21 years explained the need to solve problems, such as lack of parking space for electric kickboards, security issues, and illegal disposal of trash.

**Figure 6 ijerph-19-15834-f006:**
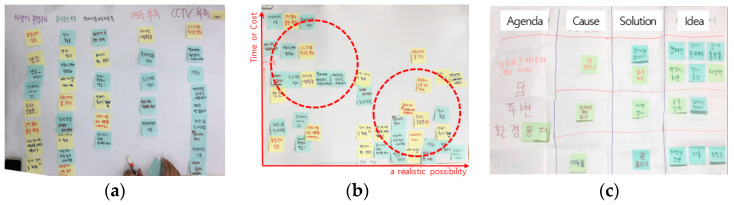
Developing ideas through the impact–effort matrix: (**a**) Producing solutions for the causes of reported problems: sticking post-its containing solutions to illegal parking, lack of rest area, illegal trash disposal, lack of streetlights, and lack of CCTV cameras. (**b**) Producing feasible solutions: grouping solutions that require less time and cost, and are feasible. (**c**) Specific ideas about feasible solutions.

**Figure 7 ijerph-19-15834-f007:**
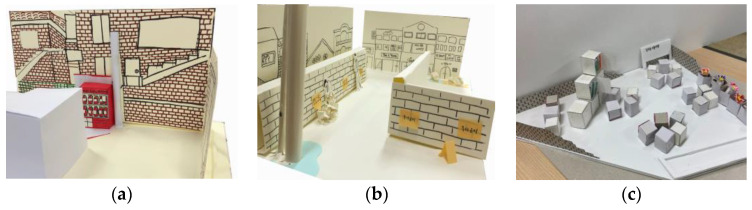
Results of idea modelling: (**a**) design of a fire extinguisher cabinet in the alley; (**b**) design of the signs on garbage disposal; (**c**) design of rest area.

**Figure 8 ijerph-19-15834-f008:**
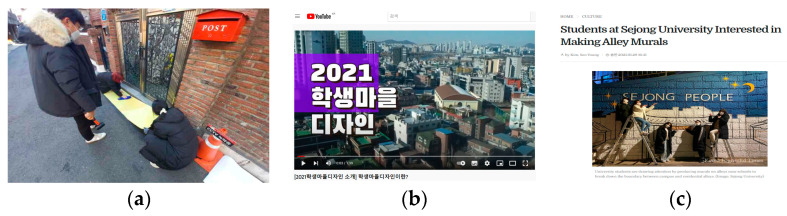
Results of project implementation: (**a**) project construction site: painting the front of resident gates to prevent the speeding of motorcycles in alleys; (**b**) releasing a video about the project outcomes (text on the screen is ‘2021 Student Village Design’); (**c**) press release.

**Table 1 ijerph-19-15834-t001:** Methods of each activity phase for residents and students when communicating and applying public design.

Phase	Methods	Main Content	Application of Public Design	Student Participation
Planning	Action planning events	Establishing activity plans	○	×
	Briefing workshops	Establishing agendas and outlines	○	○
	Field workshops	Creating local communities (developing countries)	×	×
	Microplanning workshops	Planning development plans (developing countries)	×	×
Discussion	Activity weeks	One week’s worth of theme-based activities to attract public attention	○	○
	Art workshops	Developing creativity through art making	○	○
	Community planning forums	Public forums for interactions among interested parties	○	×
	Design games	Putting together puzzles to view the layout options	○	○
	Design workshops	Developing ideas	○	○
	Gaming	Understanding other people’s views through games	○	○
	Planning days	One-day activities to explore and discuss the neighborhood and surrounding city	○	○
	Practical for real	Anticipate the priorities of local planning	○	○
	Planning weekends	Intensive discussions conducted on weekends	○	○
	Prioritizing	Prioritizing what to do and when	○	○
	Process planning sessions	Carrying out actual action plans by inviting the public sector and investors	○	○
	Reconnaissance trips	A team comprising local parties and experts exploring the region	○	○
	Photo surveys	Discussing with environmental photos	○	○
	Risk assessments	Analyzing risk factors/vulnerability/capability of the plan	○	○
	Simulations	Reproducing actual events or activities in dramas or presentations	○	○
	Urban design studios	Education to derive designs in universities	○	×
Publicizing planning process	Future search conferences	Meetings to share future visions	×	×
	Interactive displays	People communicating and exchanging opinions through planned exhibits	○	○
	Open space workshops	Discussing topics with various people	○	×
	Road shows	Workshops, exhibitions, symposiums for making urban design plans	○	×
Survey	Choice catalogues	Surveying people’s needs with a design choice list	○	○
	Community profiling	Identifying the regional situation	○	○
	Table scheme displays	Displaying the content of the plan and gathering people’s opinions	○	○
Visual expression	Diagrams	Materials showing the concepts and step-by-step information	○	○
	Electronic maps	Digital materials used to explore the region	○	○
	Elevation montages	Materials showing the street façades by gathering building photos	○	○
	Mapping	Displaying key issues and the status on a map	○	○
	Models	Materials showing ideas in 3D models	○	○
	Participatory editing	Participating in report editing from planning to results	○	×
Assessment	Award schemes	Selecting the best ideas and activities and awarding cash prizes	○	○
	Ideas competitions	Evaluation that stimulates creative thoughts	○	○
	Review sessions	Assessment that monitors the overall process and increases initiative	○	○
Publicizing planning results	Design fests	Exhibition/presentation of the planning team’s thoughts in front of people	○	○
	Local design statements	Presenting guidelines for new regional development in a report	×	×
	Newspaper supplements	Publicity of planning details and design via newspaper	○	○
	Open house events	Delivering information, such as via exhibitions, to gauge public opinions/reactions	○	○
	Street stalls	Gauging people’s reactions by holding outdoor exhibitions	○	○
	Video soapboxes	Publicity by showing videos of planning in public spaces	○	○
Creating and using spaces	Architecture centers	A space to provide information about the local architectural design	○	○
	Community design centers	A space for community organizations	○	○
	Environment shops	Providing regional information and selling related materials	×	×
	Mobile units	Moving around and providing technologies or skills required for activities	×	×
	Neighborhood planning offices	A local base important for activities	×	×
Creating and using organizations	Design assistance teams	Experts in related fields who provide new and original views	○	○
	Development of trust	An organization to run the regional regeneration and development project	×	×
	Planning aid schemes	An organization of experts providing advice on the plan	○	○
	Task forces	Local governments and academic institutions gathered to find solutions	×	×
	User groups	Members that can participate in planning from the users’ perspective	○	○
Budget	Feasibility funds	Funding required to investigate project feasibility	×	×

Note: ○ represents that an item is possible, while × indicates that it is not possible.

**Table 2 ijerph-19-15834-t002:** Analyzing each activity phase from basic survey and setting to selecting Student Village Designers.

Category		Basic Survey and Setting				Selecting Student Village Designers	
Main content	First Student Village Design	Situation survey	Deriving key issues	Using the publicity base	-	Online recruitment notice	Selecting Student Village Designers
	Second Student Village Design	-	Deriving key issues again	(Same as above)	Online communication	(Same as above)	(Same as above)
Activity phase		Survey	Visual expression	Creating and using space	Publicity	Publicity	Planning
Method		Community profiling	Mapping	Community design center	Interactive displays	Interactive displays	Briefing workshop
Operating program		On-site survey and resident opinion poll	Mapping survey results	Operating the support center, installing resident opinion boxes	Operating a YouTube channel, QR code publicity	Using the website, SNS, and YouTube channel	Holding the launching ceremony

**Table 3 ijerph-19-15834-t003:** Analyzing each activity phase of workshops.

Category		Workshops					
Main content	First Student Village Design	Deriving regional agendas (topics)	Surveying and identifying regional problems		Deriving the cause of each problem as well as solutions	Design conception	-
	Second Student Village Design	(Same as above)	(Same as above)		(Same as above)	(Same as above)	Expert advice
Activity phase		Discussion	Survey	Discussion	Discussion	Discussion, Visual expression	Assessment
Method		Art workshop	Choice catalogue	Simulations	Practical planning	Design workshop, Models	Review session
Operating program		Detailed picture generation	Resident opinion poll (sticking stickers)	Persona-based Scenario Method	Impact–Effort Matrix	Modeling	Presentation and monitoring

**Table 4 ijerph-19-15834-t004:** Analyzing each activity phase from preparing and supplementing the action plan to project implementation.

Category		Preparing and Supplementing the Action Plan			Project Implementation		
Main content	First Student Village Design	Presentations and critiques	Expert mentoring	Discussion with parties concerned	Construction	Outcome assessment	Performance sharing session
	Second Student Village Design	-	(Same as above)	(Same as above)	(Same as above)	(Same as above)	(Same as above)
Activity phase		Creating and using organizations		Creating and using organizations	Publicity	Assessment	Publicity
Method		Design assistance team		User group	Street stalls	Award scheme	Newspaper supplement,
							video soapbox
Operating program		Presentation and critique, expert mentoring		Briefing sessions, opinion polls, receiving consent, etc.	Hanging nameplates, inviting visits and participation (gift)	Presenting certificate and cash prize	Press release, videomaking, and release, etc.

**Table 5 ijerph-19-15834-t005:** Results of the effect analysis according to community planning application methods.

Process	Main Content	Community Planning Application Method	Resident Communication Method	Effect of Resident Communication	Performance-Based Effect
Basic survey and setting	Situation survey	Community profiling	Resident opinion poll	Direct effect	Related to selection of regional agendas
	Deriving key issues	Mapping	(No)	Indirect effect	Related to decision of project site
	Publicity base	Community design center	Offline resident communication base	Indirect effect	Inadequate use
	Online publicity	Interactive display	Online communication and publicity platform	Direct effect	Related to collection of resident opinions
Selecting Student Village Designers	Online recruitment notice	Interactive display	(No)	None	Used to recruit students from other universities
	Selection	Briefing workshop	(No)	None	Related to selection of students actively participating in the project
Workshop	Deriving regional agendas (topics)	Art workshop	(No)	Indirect effect	Deriving regional agendas (topics) and using them in creative activities
	Surveying and identifying regional problems	Choice catalogue	Resident opinion poll	Direct effect	Related to deriving real-life problems and their causes
		Simulation	(No)	Indirect effect	Used to identify the causes of the problems and understanding residents
	Causes and proposed solutions of the identified problems	Practical planning	(No)	Indirect effect	Related to deriving realistic solutions
	Design conception	Design workshop, Models	(No)	Communication needed	Related to specific project decisions
		Review session	(No)	Communication needed	Used to improve practicality
Preparing and supplementing the action plan	Critique and mentoring	Design assistance team	Resident representative gathering opinions	Indirect effect	Used to supplement the design
	Discussion with parties concerned	User group	Briefing session, discussion, etc.	Direct effect	Related to deciding whether to implement or not
Project implementation	After construction	Street stall	Offline publicity	Indirect effect	Used to spread positive awareness and recover trust
	Outcome assessment	Award scheme	Resident representative participating in evaluation	Indirect effect	Used to promote competition when generating effective ideas
	Performance sharing event	Newspaper supplement, video soapbox	Online publicity	Indirect effect	Used in national publicity

## Data Availability

Data can be made available upon contacting the authors.

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
