# Peer review of "Urban Regeneration Involving Communication between University Students and Residents: A Case Study on the Student Village Design Project"

_ijerph, 2022, doi:10.3390/ijerph192315834_

Round 1

Reviewer 1 Report

kindly check the uploaded file.

Reviewer 2 Report

The overall topic of the paper is interesting because it shows the collaboration between students and the local community.

Proofreading is required, as well as revision of the text and tables.

Abstract: The main topic of the research and methodology should be presented in a clear manner and should be focused on the particular topic. Start by explaining the specific conflicts regarding urban regeneration in the University towns, as the research problem.

Also, you should clearly state the specific aims of this research at the beginning of the paper. 

Section 1.2. Literature review should not include an explanation of the methodology that was used for this research, this also refers to Table 1.

The connection between the methodology explained, and the results presented should be revised. 

It is unclear how each of the different types of results are connected to the particular aims of this research. Consider reducing and focusing the results presented. 

Table 2. is hard to follow; it should be restructured and explained more/discussed in the text. 

The discussion and conclusion sections are well structured covering the implications of this research with previous studies, as well as research limitations and future research possibilities. 

Round 2

Reviewer 2 Report

Dear authors, thank you for addressing all my remarks considering your manuscript. It would be beneficial if you referenced previous or present similar research more in the theoretical background as well as in the Discussion section. Also, you presented methodology through the specific phases and processes of the project/collaboration you are describing in the manuscript. However, you should add or emphasize the scientific methodology that was used (and is usually used) for the scientific research papers.
